# Behavioral factors predict all-cause mortality in female coronary patients and healthy controls over 26 years – a prospective secondary analysis of the Stockholm Female Coronary Risk Study

**Hans-Christian Deter**[1]*, **Reinhard Meister**[2], **Constanze Leineweber**[3], **Göran Kecklund**[3], **Lukas Lohse**[4], **Kristina Orth-Gomér**[5†], **Fem-Cor-Risk Study group**[¶]

1 Medical Clinic, Psychosomatics, Charité, Campus Benjamin Franklin and German Center of Cardiovascular Research, Berlin, Germany, 2 Faculty II, Mathematics, Physics, Chemistry, Berliner Hochschule für Technik Berlin—University of Applied Sciences, Berlin, Germany, 3 Department of Psychology, Stress Research Institute, Stockholm University, Stockholm, Sweden, 4 Clinical Pharmacology, Charité, Berlin, Germany, 5 Clinical Neuroscience, Karolinska Institutet, Stockholm, Sweden

† Deceased.
¶ The complete membership of the author group "Fem-Cor-Risk Study group" can be found in the Acknowledgments
* deter@charite.de

**Data Availability Statement:** Given restrictions from the ethical review board and considering that

## Abstract

### Objective

The prognosis of coronary artery disease (CAD) is related to its severity and cardiovascular risk factors in both sexes. In women, social isolation, marital stress, sedentary lifestyle and depression predicted CAD progression and outcome within 3 to 5 years. We hypothesised that these behavioral factors would still be associated with all-cause mortality in female patients after 26 years.

### Methods

We examined 292 patients with CAD and 300 healthy controls (mean age of 56 ± 7 y) within the Fem-Cor-Risk-Study at baseline. Their cardiac, behavioral, and psychosocial risk profiles, exercise, smoking, and dietary habits were assessed using standardized procedures. Physiological characteristics included a full lipid profile, the coagulation cascade and autonomic dysfunction (heart rate variability, HRV). A new exploratory analysis using machine-learning algorithms compared the effects of social and behavioral mechanisms with standard risk factors. Results: All-cause mortality records were completed in 286 (97.9%) patients and 299 (99.7%) healthy women. During a median follow-up of 26 years, 158 (55.2%) patients and 101 (33.9%) matched healthy controls died. The annualized mortality rate was 2.1% and 1.3%, respectively. After controlling for all available confounders, behavioral predictors of survival in patients were social integration (HR 0.99, 95% CI 0.99–1.0) and physical activity (HR 0.54, 95% CI 0.37–0.79). Smoking acted as a predictor of all-

sensitive personal data are handled, it is not possible to make the data freely available. Access to the data may be provided to other researchers in line with German law and after consultation with the Charité legal department. Requests for data, stored at the Charité, Medical Clinic, Division of Psychosomatic Medicine, should be sent to registratory-cto@charite.de with reference to 'FemCorRisk – 25 years follow-up of mortality', or directly to the corresponding author.

**Funding:** K.O-G. - grant HL-45785 from the U.S. National Institutes of Health, https://www.nih.gov/ - grant 98-0336 of the Swedish Council for Work Life Research,https://forte.se/en/ - grant from the Swedish Medical Research Council, https://www.vr.se - grant of the Swedish Labor Market Insurance Company, https://www.afaforsakring.se - grant of Osher Foundation, Karolinska Institute, Stockholm. R.M. - Grant:ME 910/2-1. German Research Foundation (DFG) https://www.dfg.de/en/ H.C.D. - No:81X2300152, German Center of Cardiovascular Research. https://dzhk.de/en/partner-sites/berlin The funders had no role in study design, data collection and analysis, decision to publish, or preparation of the manuscript.

**Competing interests:** The authors have declared that no competing interests exist.

**Abbreviations:** ACS, acute coronary syndrome; AMI, acute myocardial infarction; AP, alkaline phosphatase; ALT, alanine-aminotransferase; AMI, acute myocardial infarction; AT, attachment; BMI, body mass index; CCS, chronic coronary syndrome; CI, confidence interval; DHEAS, dehydroepiandrosterone sulfate; ECG, electrocardiogram; Fem-Cor-Risk, Stockholm Female Coronary Risk Study; HRT, hormone replacement therapy; HRV, heart rate variability; HS, household size; LV, left ventricular; NSTEMI, non-ST-elevation myocardial infarction; MACE, major adverse coronary event; MI, myocardial infarction; PA, physical activity; UAP, unstable angina pectoris; Urat, uric acid; SDNN, standard deviation of the normal-to-normal R-R interval; SES, socio economic status; SI, social integration; SMSS, Stockholm Marital Stress Scale; STEMI, ST-elevation myocardial infarction; VE, vital exhaustion.

cause mortality (HR 1.56, 95% CI 1.03–2.36). Among healthy women, moderate physical activity (HR 0.42, 95% CI 0.24–0.74) and complete HRV recordings (≥50%) were found to be significant predictors of survival.

## Conclusions

CAD patients with adequate social integration, who do not smoke and are physically active, have a favorable long-term prognosis. The exact survival times confirm that behavioral risk factors are associated with all-cause mortality in female CAD patients and healthy controls.

## Introduction

Cardiovascular disease remains the leading cause of death among women worldwide. Growing knowledge, technical advances in cardiology and increasing interest in including women with coronary artery disease (CAD) in clinical trials have led to better recognition of the risk factors for developing CAD and adverse outcomes in this group [1]. In a 19-year follow-up study of healthy subjects, fasting plasma triglyceride levels, age, systolic blood pressure, smoking and, in men only, plasma cholesterol were found to be independent risk factors for death from myocardial infarction (MI) [2]. In women with CAD, acute myocardial infarction (AMI) as an index event, left ventricular dysfunction and diabetes mellitus were shown to be additional risk factors [3]. Social risk factors, such as low education level, predicted the 20-year incidence of MI or coronary death in healthy women after controlling for cardiovascular risk factors [4].

If looking at a comprehensive analysis of factors contributing- to the development and progression of CAD, the Stockholm Female Coronary Risk Study (Fem-Cor-Risk-study), a population-based study of women with CAD and age-matched healthy controls [5], opens new vistas on different social and behavioral risk factors associated with all-cause mortality outcomes: Work stress [6,7] and family stress [8–10] influence the prognosis of patients after AMI and unstable angina pectoris (UAP) [11]. Socioeconomic status (SES) [12], lack of social integration [13] and social isolation [14,15] were determinants of hard CAD endpoints or progression to coronary atherosclerosis after the first cardiac event. Depression and exhaustion were found to be important psychological risk factors [15–20].

Interactions of these psychosocial parameters with biological risk factors such as lipids [1], inflammation [21], autonomic imbalances [22,23], coagulation [24], alcohol [25] and sleep disorders [26] exacerbate the progression of coronary atherosclerosis and trigger cardiovascular events. Behavioral factors such as physical activity (PA) [27], smoking [28] and dietary habits [21] interact with social and psychological [29] dimensions and with comorbidities such as diabetes [3] and hypertension.

The influence of behavioral and psychological factors on cardiac risk factors has also been demonstrated in healthy populations. Interactions between social factors and psychological and biochemical dimensions have been shown cross-sectionally [30,31] and in a nine-year follow-up study in healthy controls from the Fem-Cor-Risk study [32]. The importance of social relationships on mortality risk was demonstrated in a meta-analytic review of 148 studies [33].

The main objective of the Fem-Cor-Risk-study was to examine whether psychosocial strain combined with unhealthy lifestyle habits contribute to the development and progression of CAD in younger women.—Cross-sectional and 3 three-year, 5 five-year and in 2 nine-year prospective follow-up studies with FemCorRisk patients have largely confirmed that lack of social integration [15,30], received stress [8,10] and behavioral risk factors as sedentary lifestyle

[27], depression/vital exhaustion [15,19,20] and sleep disturbances [26] are predictors of increasing coronary stenosis, recurrent cardiac events, cardiac and all-cause mortality (S1 Table). Psychobiological pathogenetic mechanisms [21–26] and aspects of the female reproductive system have been implicated in this interaction.

In this exploratory analysis using data from the Fem-Cor-Risk study, we examined the hypothesis that social and behavioral factors are associated with all-cause mortality at a very long follow-up. We hypothesised that, in addition to CAD severity, social strain (lack of social integration and marital and work stress experiences) and behavioral risk factors (sedentary lifestyle, smoking, depressive symptoms/vital exhaustion and sleep disturbances) would be associated with all-cause mortality at 26 years in an unselected sample of women with CAD.

In addition, we hypothesised that these psychosocial factors would have an impact on all-cause mortality in healthy controls after 26 years of follow-up.

## Methods

### Design and subjects

We examined all-cause mortality in a prospective cohort study with randomly selected, age matched controls of women 65 years of age or younger who had lived in the greater Stockholm area (5). All female patients were hospitalized for an acute CAD event (110 (37%) AMI, 182 (63%) UAP) between February 1991 and February 1994. Controls were randomly selected from the city census and were matched by age and catchment area. Age at entry (mean, SD) was 56 ± 7 years and equal in the patient (n = 292) and control (n = 300) cohort by matching.

The final available update of mortality status of participants (collection time between 15 September 2019 and 10 February 2020) linking the unique 10-digit person identification numbers to the community health care registers provided a follow-up time median 26.1 years, range 25.3–28.3 years. The status update was successful in 286 (97.9%) of 292 patients and 299 (99.7%) of 300 controls. Seven subjects moved away or were not registered in Sweden.

### Data sources

The dataset of the Fem-Cor-Risk Study contained 292 female patients with CAD from Stockholm, Sweden, and 300 matched healthy female controls. Data about 160 variables were obtained from medical examination and survey responses shortly after study entry [5,11]. For this study, we examined two main criteria related to social strain and behavioral risk factors.

### Social strain

a. Lack of social integration
   For the measurement a modified version of the Interview Schedule for Social Interaction with two main dimensions was used [14,15,29,34]: *Social Integration (SI)* measured the relationships to friends, neighbors, work associates and acquaintances. This includes a sense of belonging, practical help and support. *Attachment (AT)* measured close emotional ties.

b. Stress experience

   • *Marital stress* was measured by a structured interview using the Stockholm Marital Stress Scale [8]. One point was given for each question answered in a way that reported stress, and a point was given for every problem reported (infidelity, substance use/abuse, economic problems, health problems, or other unspecific problems). Higher scores indicate higher marital stress.

- *Work-stress* was measured using the Swedish version of the Karasek demand–control questionnaire, which has been tested for consistency and reliability in the Swedish population [35]. Strain at work was derived from the ratio of psychosocial job demands to control. A large ratio indicated increased job stress, which occurs when demands exceed control.

## Behavioral risk factors

a. *Physical activity* (PA) was assessed according to the World Health Organization criteria and graded into (I) reading, watching TV, or other sedentary leisure activities; (II) walking, cycling, or other forms of physical activity; (III) exercises to keep fit, heavy gardening, etc., for at least 4 h a week; and (IV) hard training or participation in competitive sports several times a week [36]. In the analyses, physical exercise in leisure time was dichotomized into sedentary lifestyle (I) and non-sedentary lifestyle (II, III and IV). Q-shaped patterns (I and III/IV vs. II) and linear (L-) patterns (I vs. II vs.III/IV) were examined separately. In an additional measurement, patients reported their present physical activity (posture) during the 24 h ECG measurement at leisure time 7 times a day (apply ECG, morning, after lunch, afternoon, after dinner, before sleeping, awakening). A measurement of physical activity in the morning and after lunch was specified as "Posture_d2".

b. *Smoking status* was categorized as current, or non and previous smoker (1 year prior to the date of study recruitment).

c. *Depressive symptoms* were assessed using a short and convenient self-rating questionnaire, originally developed by Pearlin [37]. Nine common symptoms of depressive feelings are presented; the recent experience (preceding month) of each symptom is checked, and the number of symptoms is added to a score from 0 to 9. The scale had an adequate internal consistency (Cronbach's alpha = 0·85) and was significantly correlated (r = 0.70; p < .0001) with the Beck Depression Inventory in a pilot study of Swedish women [11]. *Vital exhaustion* (VE) was measured by an earlier version of the Maastricht Questionnaire [11,20]. The scale yields 18 items and includes questions on fatigue, irritability, depressed affect, and personal accomplishment. The scale demonstrated excellent internal reliability (Cronbach's alpha 0.93).

d. *Disturbed sleep* was measured by Karolinska Sleep Questionnaire (KSQ) [26,38]: The women were asked about mental fatigue, not feeling well-rested on awakening, disturbed or restless sleep, premature awakening, and difficulties falling asleep. The response alternatives were "never," "rarely," "some of the time," and "most of the time." A disturbed sleep index was computed by simply adding the item scores (range 0 to 15). A higher score indicated more frequently disturbed sleep. Additionally, we used a variable, *sleep quality*, in women who responded with "some of the time" or "most of the time. These women received 1 score point in each of three sleep questions: disturbed sleep, premature awakening and difficulties falling asleep. Patients who responded with "never" or "rarely" received no points in this examination (range 0 to 3). Cronbach's alpha for the sleep quality index of the KSQ was reported to be .80 [38].

*Covariates*: Age at baseline examination, body mass index, menopausal status and educational level were assessed by standardized methods (S1 File). The severity of heart failure (Killip classification), systolic blood pressure, serum levels of cholesterol, HDL, and triglycerides, as well as CRP, coagulation factors, uric acid, alkaline phosphatase, AST, ALT and DHEAS, were assessed at clinical examination.

Coronary angiography and further measurements of the study related to left ventricular dysfunction are described in Supplemental material, S1 File.

In the ECG 24-h-HRV assessment, we included only HRV recordings >50% ("HRV-filter") [22]. ECG recordings were excluded if they showed more than 10% nonsinus rhythm (seven patients), or less than 50% of the original ECG recording was available for analysis (eight patients) due to technical artifacts, atrial fibrillation or frequent arrhythmia.

Further psychological questionnaire data related to anger, hostility, and self-esteem were included in the Cox Boost model and described in the Supplemental material, S1 File.

Data were recorded in an updated dataset of Fem-Cor-Risk-study [5] with a follow-up of median 26.1 (range 25.3–28.3) years. The control cohort was analyzed separately using the same model-building strategy without cardiac markers of CAD and findings of coronarangiography.

The only endpoint was survival time characterized by all-cause mortality.

## Data analysis

The first step of our analysis shows survival curves—all-cause mortality—of patients and controls. The differences in estimated survival probability are obvious, with different shapes for the two groups. The density plot of the matched ages shows agreement between patients and controls. We opted to reject the idea of a common model for analyzing risk factors, as controls and patients are different. In addition, several cardiological markers of CAD are not available for healthy controls.

Our main interest was the question of whether baseline psychosocial status might be among well-established clinical conditions predictors of all-cause mortality, even after a very long follow-up period of up to 26 years. The complexity of estimable logistic regression-based risk factors of survival rates in given periods, like 5-, 10-, and 15-year survival, is limited. Therefore, different models have been applied to the same data in the past [8]. Using individual survival times for a regression approach would allow assessing the effects of many potential risk factors simultaneously. Variable selection is mandatory, given the very rich set of baseline covariates of more than 80 per individual.

We used a method for boosted regression (COX Boost) [39] for variable selection, arriving at a sparse pragmatically constructed prediction model of survival (S2 File).

In an additional explorative analysis, a subgroup of patients stratified by age and social integration was analyzed further. Mortality predictors were calculated after 15 years, ten years after behavioral follow-up data were published by the Fem-Cor-Risk-Study [8,11]. A different impact of these predictors in the Cox Boost analysis could show the strengthening or weakening of the influence of a baseline predictor within the observation time of 15 to 26 years.

**Descriptive statistics.** The distribution and characteristics of all covariates, including all variables related to the hypotheses to be elucidated, are given in Supplement 1. Splitting the study cohort into long-term survivors (t > 25 years) and earlier deaths generates a retrospective view. Information about the baseline status of long-term survivors may be of interest. Disentangling the role of psychosocial and behavioral indicators on the one hand and classical clinical covariates on the other hand under the influence of unknown comorbidities is a challenge. Therefore, we choose an exploratory, pragmatic approach for building a model of all available covariates. We did not provide formal testing between patients who survived and died, avoiding misinterpretation of apparent relations.

**Regression analysis.** The analysis plan for this study is based on a multiple regression approach for survival times using explorative modeling to generate an interpretable prediction model. Due to the long follow-up period, we have a substantial number of uncensored cases;

censoring occurs only for long-term survivors or loss to follow-up cases. This fact enables the use of multiple regression, being mandatory for unbiased results if risk factors and other covariates are unbalanced [40].

**Variable selection.** After preselecting variables in a two-step approach, we used a machine-learning technique for model building (S2 File).

*Setting up a regression model.* Choosing one of the new machine-learning approaches, the Cox Boost concept [39] (https://GitHub.com/binderh/CoxBoost), which has been used and cited by many authors from various fields of medical research, such as cancer or cardiology [41], we can overcome the **p** $>$ = **e** dilemma.

A special feature of the algorithm based on penalized ridge estimation is the possibility of including unpenalized candidate variables. In this way, we could select our hypothesis-related psychosocial and behavioral variables with higher priority. Originally, Binder and Schumacher [39] tried eliminating known clinical predictors for survival when investigating the influence of molecular markers in cancer studies. We are in a reverse setting: we want to learn whether our hypotheses hold in a multiple regression model, including the most relevant clinical predictors of survival.

Therefore, our final analysis is based on all variables selected by Cox Boost as input for a traditional Cox regression estimation, without "privileges" for the candidate variables. Four behavioral and social hypothesized variables (sleep disturbances, depression, work strain and social integration) were added to the fifteen variables selected by Cox Boost. In the sub-cohort of women with partners, we calculated the impact of marital stress on the outcome and added it as a variable, comparing this variable with predictors of survival in the final model.

Tests of the proportional hazard assumption are applied for model checking. We handled technical aspects of data cleaning, transformations, recoding of categorical variables, log transformation of clinical chemistry, imputation of missing values and finally the tuning of the fitting algorithm required to avoid overfitting and spurious results due to leverages of the covariate data (S3 File).

All these aspects are documented in a technical report, providing reproducible research standards via R (version 3.6.3) (available on request by the authors).

Results are presented in tables, plots of hazard ratios and predicted survival curves. It is important not to ignore un-significant results in an exploratory analysis that pragmatically examines known and new hypotheses. Concerning the role and interpretation of the model variables we emphasize, that (i) the direction of the estimated HR can be checked for plausibility, (ii) the size of the estimated effect should be checked for relevance, and (iii) confidence intervals address the question of statistical significance. All these aspects of the estimated hazard factors are regarded and reported in the result.

*Ethics.* Patients and healthy control subjects who agreed to participate in the study after being informed verbally and in writing by a study nurse signed an informed consent form. Furthermore, the confidentiality of the information and the right to terminate the study if necessary were confirmed.

The study was approved by the ethics committees of the *Karolinska Hospital*, Stockholm (91;119 and 02;202), and the *Charité Universitätsmedizin*, Berlin (EA4/168/19).

## Results

Samples: Of the 292 patients 238 patients had an angiogram and left ventricular angiography. 22 (9.2%) showed a non-obstructive CAD (AMI with a normal angiogram or minimal coronary lesions), and 66 (27.7%) patients a mild to moderate coronary atherosclerosis (30% to <50%). Sixty-two (26%) of patients had one-vessel disease, 36 (15%) had two-vessel disease

and 53 (22%) had three-vessel disease (including nine patients with significant left main stenosis) [3].

The 300 age matched presumably healthy control subjects (S1 Fig) had no symptoms of heart disease and had not been hospitalised for any disease in the previous 5 years. Of the women studied, 74 patients (25%) and 84 control subjects (29%) were premenopausal.

Twenty-six years (range 25.3–28.3) after the first coronary event, 128 (44.8%) of women with CAD were alive, and 158 (55.2%) had died (six patients (2.1%) we did not reach). Of the age-matched presumably healthy controls, 198 (66.2%) were alive, and 101 (33.8%) had died (one women (0.3%) we did not reach). During the follow-up evaluations, the proportion of deceased women in the patient group compared to the control group decreased from a factor of 9.0 at 7 years to a factor of 3.4 at 15 years and finally to a factor of 1.6. Considering the time points for 75% survival we found a time shift of 8.4 years (95% CI 4.2–12.6) in favor of controls (Fig 1 and S2 Table). In our study, 68% (n = 200) of the patients were working at the time of

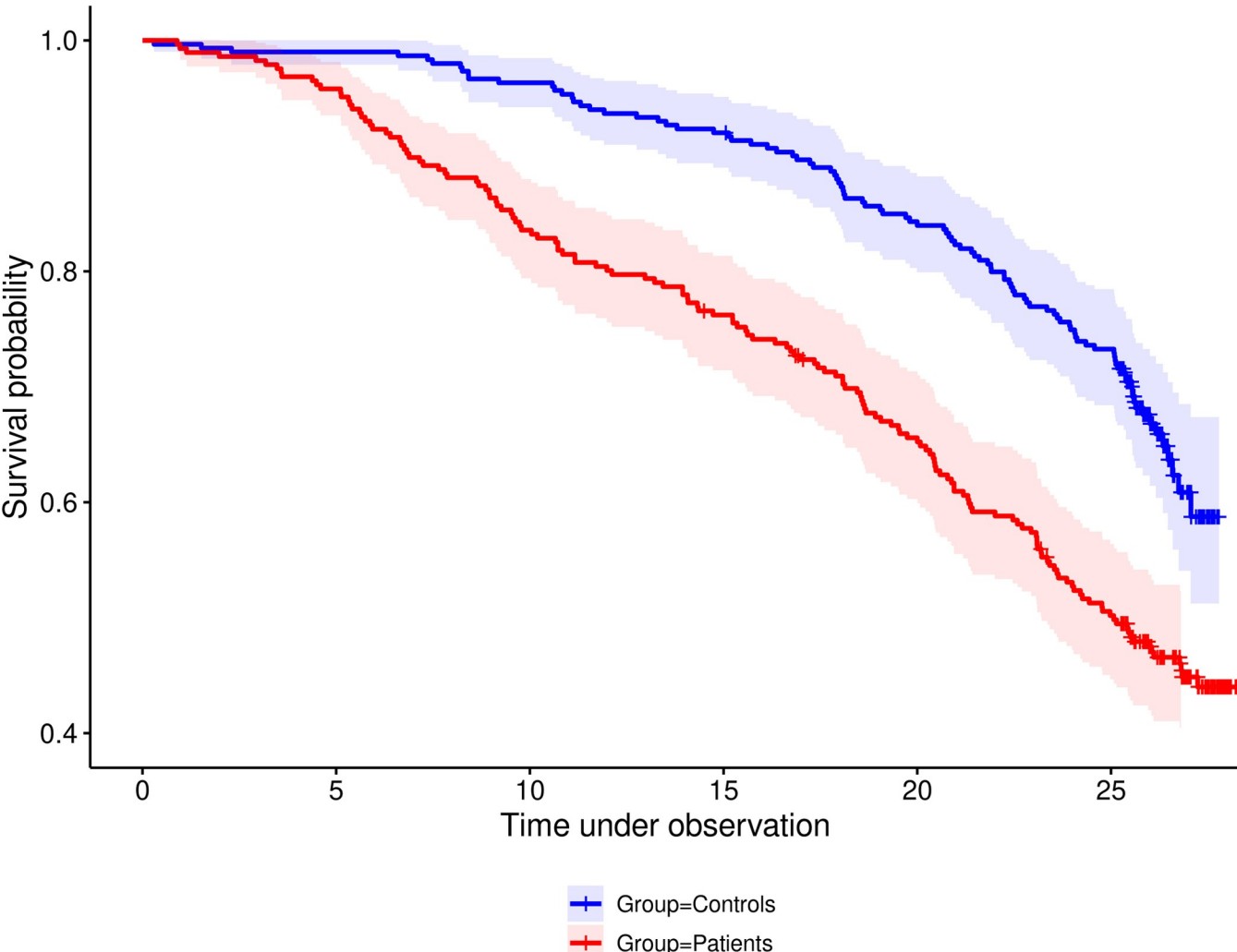

**Fig 1. Survival in female CAD patients and healthy controls.** Kaplan Meier estimates including 95% point wise CIs are displayed in red for patients and blue for controls. The difference is quite obvious. The chance for surviving 26 years of follow up is below 50% for patients, compared to almost 60% of healthy controls. Considering the time points for 75% survival we find a time shift of 8.4 years (approx. 95% CI 4.2–12.6) in favor of controls.

examination compared to 83% (n = 249) of the healthy cohort. Nearly two-thirds of the 299 healthy controls and the 292 patients (187, 64%) were married or cohabiting with a partner.

The distribution of clinical, social, and psychological characteristics of 286 female CAD patients and healthy controls at baseline and survival at the 26-year follow-up is presented as proportions or median and range in Tables 1 and 2. Patients who survived were less often highly educated and more often married. Physical activity and not smoking showed individual advantages for survival, whereas serious depression or exhaustion did not. Most standard risk factors (age, BMI, waist–hip ratio, triglyceride, cholesterol, HDL, LDL) and comorbidities (left ventricular dysfunction, diabetes) indicated in this individual evaluation had associations to all-cause mortality.

In healthy controls, marital status, education, physical activity and smoking were similar behavioral predictors to BMI, cholesterol, systolic blood pressure (SBP) and heart rate variability (HRV). The present statistical procedure allowed examining how many of the hypothesized variables were selected to be among the best 20 predictors and reached sufficient predictor quality in the time between baseline and follow-up. Any of the variables that reached a high predictor quality summarized in each of the calculated 26 years were included in that model. Non-hypothesized predictors could also be included in this exploratory analysis. The results of this calculation are seen in Table 3. In this model, 19 variables and the global score showed no restrictions (proportionality; Schoenfeld test; S3 Table).

## A. Related to our hypotheses, the following results emerged in CAD patients

1. Social strain

- *Lack of social integration* and individual survival or mortality over 26 years are presented in Table 3A. We found an interaction between SI and age upon study entry (HR = 0.99, 95% CI 0.99–1.0, p = 0.042). The hypothesis related to lack of social integration seems to be supported by this analysis: older women died if they were socially not integrated. However, we found that younger women survived if they had less social integration (Fig 2). These results were only partly in line with our hypothesis: Patients with age around 65 and a lack of social integration (score = 10) showed a mortality of 63%, but patients aged around 45 with low social integration (score = 10) showed a low mortality (11%; Figs 2 and S2). The situation we found was much more differentiated than hypothesized. The variables attachment and household size were not selected in the model.

| Age | Social_Integ | 26 year mortality | 95% lower | 95% upper |
|---|---|---|---|---|
| 45 | 10 | 11% | 4% | 25% |
| 65 | 10 | 65% | 43% | 86% |
| 45 | 30 | 37% | 19% | 65% |
| 65 | 30 | 37% | 23% | 57% |

- Stress experience
  Stress was no significant predictor for all-cause mortality at the 26-year follow-up:

  a. Marital stress was higher in patients with all-cause mortality. Due to missing values of patients without a partner, this variable was adjusted by comparing its regression with the

**Table 1. Clinical, social and psychological characteristics of 286 female CAD patients at baseline and survival at 26 years follow-up.**

| | All patients, n (%) | Survivors, n (%)* | Nonsurvivors, n (%)* | Effect size η²** |
|---|---|---|---|---|
| | 286 (100) | 133 (46.5) | 153 (53.5) | |
| Marital status: single | 24 (8.5) | 8 (6.1) | 16 (10.6) | |
| widowed | 19 (6.7) | 6 (4.5) | 13 (8.6) | .02 |
| divorced | 60 (21.2) | 26 (19.7) | 34 (22.5) | |
| married | 180 (63.6) | 92 (69.7) | 88 (58.3) | |
| Education: Mandatory | 179 (62.6) | 90 (67.7) | 89 (58.2) | .01 |
| high school + college/university | 107 (37.4) | 43 (32.3) | 64 (41.8) | |
| Marital stress: no | 110 (59,8) | 60 (66.7) | 50 (51.1) | .02 |
| yes (>3, upper 2 quartiles) | 74 (40,2) | 30 (33.3) | 44 (48.9) | |
| Work stress: no | 106 (37.1) | 58 (43.6) | 48 (31.3) | |
| yes (> 0,73, upper 2 quartiles) | 93 (32.5) | 51 (38.3) | 42 (27.5) | .09 |
| homemakers | 87 (30.4) | 24 (18.0) | 63 (41.2) | |
| Diagnosis at index event | | | | |
| AMI | 107 (37.4) | 43 (32.3) | 64 (41.8) | .01 |
| UAP | 179 (62.6) | 90 (67.7) | 89 (58.2) | |
| Menopausal status Premenopausal | 62 (21.7) | 41 (30.8) | 21 (13.7) | |
| Postmenopausal with HRT | 33 (11.5) | 23 (17.3) | 10 (6.5) | .09 |
| Postmenopaual without HRT | 191 (66.8) | 69 (51.9) | 122 (79.7) | |
| Cigarette smoking | | | | |
| Nonsmokers | 92 (32) | 54(40.6) | 37 (24.2) | .03 |
| Previous smokers | 140 (48) | 59 (44.4) | 79 (51.6) | |
| Current smokers | 58 (20) | 20 (15.0) | 37 (24.2) | |
| Physical activity: Sedentary lifestyle | 70 (24.5) | 21 (15.9) | 49 (31.8) | .04 |
| Moderate exercise | 202 (70.6) | 105 (79.5) | 97 (63.0) | |
| Regular intensive exercise | 14 (4.9) | 6 (4.5) | 8 (5.2) | |
| Left ventricular dysfunction | 30 (14) | 2 (6.7) | 28 (93.3) | .11 |
| Use of b-blockers | 197 (63.6) | 84 (46.2) | 98 (53.2) | .0 |
| Use of statins | 23 (8) | 9 (39.1) | 14 (60.9) | .0 |
| History of hypertension | 144 (50) | 60 (42.6) | 81 (57.4) | .01 |
| History of diabetes mellitus | 33 (11) | 13 (31.7) | 28 (68.3) | .01 |
| Family history CHD | 154 (54) | 68 (44.2) | 86 (55.8) | .0 |
| | Median (range) | Median (range) | Median (range) | |
| Age (years) | 57 (30–66) | 53 (30–66) | 60 (39–66) | 0.11 |
| BMI (kg m$^{-2}$) | 26.6 (18.6–44.1) | 26.2 (18.6–37.9) | 27 (18.8–44.1) | 0.02 |
| Waist–hip ratio | 0.84 (0.65–1.25) | 0.83 (0.65–1) | 0.86 (0.69–1.25) | 0.07 |
| Systolic blood pressure (mmHg) | 120 (84–180) | 118 (85–174) | 122 (84–180) | 0.02 |
| Triglycerides (mmol L$^{-1}$) | 1.4 (0.5–25.1) | 1.3 (0.5–25.1) | 1.5 (0.5–10.7) | 0.01 |
| Cholesterol (mmol L$^{-1}$) | 6.4 (3.2–11.4) | 6 (3.2–11.4) | 6.7 (3.6–10.6) | 0.05 |
| HDL (mmol L$^{-1}$) | 1.4 (0.48–3.1) | 1.48 (0.48–2.8) | 1.34 (0.72–3.1) | 0.01 |
| LDL (mmol L$^{-1}$) | 4.1 (0–8.6) | 3.8 (0–7) | 4.4 (0–8.6) | 0.02 |
| Social integration (score) | 20 (6–36) | 20 (7–36) | 21 (6–35) | .0 |
| Depressive symptoms (score) | 3 (0–9) | 3 (0–9) | 4 (0–9) | 0.01 |
| Vital exhaustion (score) | 39 (19–60) | 38 (21–60) | 40 (19–56) | 0.01 |

AMI, acute myocardial infarction; UAP, unstable angina pectoris; HRT, hormone replacement therapy; BMI, body mass index; HDL, high-density lipoprotein; LDL, low-density lipoprotein.

*Percentage of survivers or nonsurvivers in that column.

** propability estimation of predictors; η² (Cohen 1988): small effect 0.01–0.039; medium effect 0.04–0.11; large effect 0.12–0.20.

total model, which was highly significant (HR 2.66, 95% CI 2.14–3.31, p = 0.001). An HR coefficient of 1.03, 95% CI 0.97–1.1, showed a tendency with risk, but a small overlap

**Table 2. Clinical, social and psychological characteristics of 299 healthy women at baseline and survival at 26 years follow-up.**

| | All women, n (%) | Survivors, n (%)* | Nonsurvivors, n (%)* | Effect size $\eta^{2**}$ |
|---|---|---|---|---|
| | 299 (100) | 198 (66.1) | 101 (33.9%) | |
| Marital status:    single | 31 (10.5) | 21 (10.8) | 10 (10.3) | .03 |
| widowed | 24 (8.2) | 13 (6.7) | 11 (11.1) | |
| divorced | 52 (17.7) | 27 (13.8) | 25 (25.3) | |
| married | 187 (63.6) | 134 (68.7) | 53 (53.5) | |
| Education:    Mandatory | 140 (47.6) | 92 (47.4) | 48 (48) | .0 |
| high school + college/university | 154 (52.4) | 102 (52.6) | 52 (52) | |
| Menopausal status Premenopausal | 76 (26.3) | 67 (35.6) | 9 (9) | .08 |
| Postmenopausal with HRT | 92 (31.8) | 59 (29.8) | 33 (33) | |
| Postmenopaual without HRT | 121 (41.9) | 72 (36.4) | 58 (58) | |
| Cigarette smoking:    Nonsmokers | 137 (45.8) | 98 (49.2) | 39 (41.5) | .01 |
| Previous smokers | 72 (24.1) | 48 (24.1) | 24 (25.5) | |
| Current smokers | 90 (30.1) | 53 (26.6) | 31 (32.9) | |
| Physical activity:    Sedentary lifestyle | 54 (18.2) | 28 (14.2) | 26 (26.3) | .03 |
| Moderate exercise | 212 (71.6) | 152 (77.2) | 60 (60.6) | |
| Regular intensive exercise | 30 (10.1) | 17 (8.6) | 13 (13.1) | |
| Vital exhaustion quartile:    1 0–28 | 71 (25.8) | 45 (24.7) | 26 (28.0) | .01 |
| 2 29–32 | 65 (23.6) | 49 (25.3) | 16 (10.8) | |
| 3 33–37 | 68 (24.7) | 43 (23.6) | 25 (26.9) | |
| 4 >37 | 71 (25.8) | 45 (24.7) | 26 (28.0) | |
| History of hypertension | 31 (10.7) | 21 (67.7) | 10 (32.3) | .0 |
| History of diabetes mellitus | 3 (11) | 2 (31.7) | 1 (68.3) | |
| Family history of CHD | 88 (29.6) | 55 (62.5) | 33 (37.5) | .0 |
| | Median (range) | Median (range) | Median (range) | |
| Age (years) | 58 (31–67) | 54 (31–67) | 61 (42–67) | 0.13 |
| BMI (kg m$^{-2}$) | 24.6 (17.6–48.6) | 24.2 (17.6–44.4) | 25.7 (18.4–48.6) | 0.02 |
| Waist–hip ratio | 0.79 (0.53–1.44) | 0.78 (0.59–1.44) | 0.80 (0.53–0.98) | 0.01 |
| Systolic blood pressure (mmHg) | 118.5 (84–180) | 116 (86–177) | 124 (96–170) | 0.05 |
| Triglycerides (mmol L$^{-1}$) | 0.9 (0.3–4.2) | 1.6 (0.4–4.2) | 1.0 (0.3–3.2) | 0.01 |
| Cholesterol (mmol L$^{-1}$) | 6.0 (3–10.6) | 5.8 (3.0–10.6) | 6.3 (3.9–8.7) | 0.02 |
| HDL (mmol L$^{-1}$) | 1.7 (0.83–3.34) | 1.7 (0.43–3.34) | 1.8 (0.86–3.1) | .0 |
| LDL (mmol L$^{-1}$) | 3.8 (0–8.6) | 3.7 (0.8–7.6) | 4.0 (1.2–6.5) | 0.01 |
| Glucose | 4.9 (3.6–8.4) | 4.9 (3.6–7.3) | 5.0 (3.6–8.4) | 0.01 |
| Fibrinogen g/l | 3.1 (0.56–2.68) | 3.1 (2.0–5.6) | 3.3 (2.1–5.5) | 0.01 |
| v. Willebrand Factor % | 1.11 (0.43–1.13) | 1.04 (0.43–2.53) | 1.23 (0.54–3.13) | 0.03 |
| CRP | 4 (1–97) | 4 (1–28) | 4 (1–97) | 0.01 |
| AST | 0.32 (0.14–4.24) | 0.31 (0.14–0.76) | 0.35 (0.15–4.24) | 0.02 |
| SDNN | 42 (18–116) | 44 (21–80) | 38 (18–116) | 0.02 |
| VLF | 471.5 (69–1850) | 506 (86–1850) | 427.5 (69–1808) | 0.03 |
| Quality of sleep questionnaire (score) | 9.0 (0–19 | 9.0 (0–19) | 9.0 (0–17) | .0 |
| Depressive symptoms (score) | 1.0 (0–9) | 1.0 (0–9) | 1.0 (0–7) | .0 |

HRT, hormone replacement therapy; BMI, body mass index; HDL, high-density lipoprotein; LDL, low-density lipoprotein.

*Percentage of survivers or nonsurvivers in that column.

** propability estimation of predictors; $\eta^2$ (Cohen 1988): small effect 0.01–0.039; medium effect 0.04–0.11; large effect 0.12–0.20.

with no effect levels. In bivariate analyses without consideration of age, married patients had a higher survival than widowed, divorced, and single patients (Table 1; not adjusted). The small subsample of 99 patients living alone without partner and with depressive symptoms showed a high but not significant association with all-cause mortality.

**Table 3. Predictor models for women with CAD using Cox Boost analysis.**

**a. Model after 26 years**

| | Hazard-Ratio | 95% lower | 95% upper | z-value | p-value |
|---|---|---|---|---|---|
| Age | 1.181 | 1.049 | 1.331 | 2.739 | 0.006 |
| Social Integration | **1.379** | **1.011** | **1.882** | **2.030** | **0.042** |
| Social Integration x age | **0.994** | **0.989** | **1.000** | **-2.036** | **0.042** |
| Work stress (Karasek) | 0.997 | 0.963 | 1.032 | -0.171 | 0.864 |
| Practising moderate exercise* (WHO) | **0.543** | **0.373** | **0.792** | **-3.174** | **0.002** |
| Physical activity during 24 h. ECG | 0.920 | 0.840 | 1.008 | -1.793 | 0.073 |
| Current smokers | **1.559** | **1.028** | **2.363** | **2.091** | **0.037** |
| Depressive symptoms | 1.011 | 0.944 | 1.083 | 0.308 | 0.758 |
| Disturbed sleep (KSQ:>1.74) | 1.064 | 0.712 | 1.592 | 0.304 | 0.761 |
| Killip class >1 | 1.957 | 1.159 | 3.305 | 2.511 | 0.012 |
| LV dysfunction | 2.851 | 1.759 | 4.622 | 4.251 | 0.000 |
| Glucose_log | 2.634 | 1.608 | 4.315 | 3.846 | 0.000 |
| DHEAS_log | 0.673 | 0.458 | 0.987 | -2.025 | 0.043 |
| Postmenopausal without HRT | 1.436 | 0.741 | 2.782 | 1.071 | 0.284 |
| Postmenopausal with HRT | 0.895 | 0.362 | 2.212 | -0.240 | 0.810 |
| Homemaker | 1.006 | 0.690 | 1.467 | 0.033 | 0.973 |
| Uric acid_log | 1.773 | 0.868 | 3.622 | 1.572 | 0.116 |
| ALT_log | 0.664 | 0.452 | 0.975 | -2.088 | 0.037 |
| Alk P_log_sd** | 6.481 | 0.909 | 46.191 | 1.865 | 0.062 |
| Age x Alk P_ log_sd** | 0.968 | 0.937 | 1.002 | -1.870 | 0.062 |

**b. Model after 15 years**

| | Hazard-Ratio | 95% lower | 95% upper | z-value | p-value |
|---|---|---|---|---|---|
| Age | 1.232 | 1.024 | 1.483 | 2.209 | 0.027 |
| Social Integration | 1.585 | 0.963 | 2.607 | 1.813 | 0.070 |
| Social Integration x age | 0.992 | 0.983 | 1.000 | -1.898 | 0.058 |
| Work stress (Karasek) | 1.004 | 0.948 | 1.064 | 0.141 | 0.888 |
| Practising moderate exercise* (WHO) | **0.422** | **0.241** | **0.739** | **-3.022** | **0.003** |
| Physical activity during 24 h. ECG | 0.871 | 0.755 | 1.005 | -1.892 | 0.059 |
| Current smokers | 1.153 | 0.591 | 2.251 | 0.418 | 0.676 |
| Depressive symptoms | 0.962 | 0.865 | 1.069 | -0.722 | 0.471 |
| Disturbed sleep (KSQ:>1.74) | 1.309 | 0.697 | 2.456 | 0.838 | 0.402 |
| Killip class >1 | 1.383 | 0.629 | 3.040 | 0.806 | 0.420 |
| LV dysfunction | 2.544 | 1.213 | 5.336 | 2.470 | 0.014 |
| Glucose log | 3.643 | 1.934 | 6.863 | 4.001 | 0.000 |
| DHEAS_log | 0.716 | 0.420 | 1.220 | -1.228 | 0.219 |
| Postmenopausal without HRT | 1.507 | 0.486 | 4.677 | 0.710 | 0.478 |
| Postmenopausal with HRT | 1.789 | 0.449 | 7.129 | 0.825 | 0.410 |
| Homemaker | 0.848 | 0.469 | 1.534 | -0.545 | 0.586 |
| Uric acid_log | 2.730 | 0.982 | 7.590 | 1.925 | 0.054 |
| ALT log | 0.688 | 0.396 | 1.194 | -1.331 | 0.183 |
| Alk P_log_sd** | 2.160 | 0.125 | 37.210 | 0.530 | 0.596 |
| Age x Alk P_ log_sd** | 0.985 | 0.938 | 1.035 | -0.592 | 0.554 |

* Moderate and regular intensive exercise vs. sedentary lifestyle.

**x = log Alk Phos categorized according to x < mean+sd, x ∈ (mean-sd, mean +sd], x > mean+sd. This variable was standardized twice.

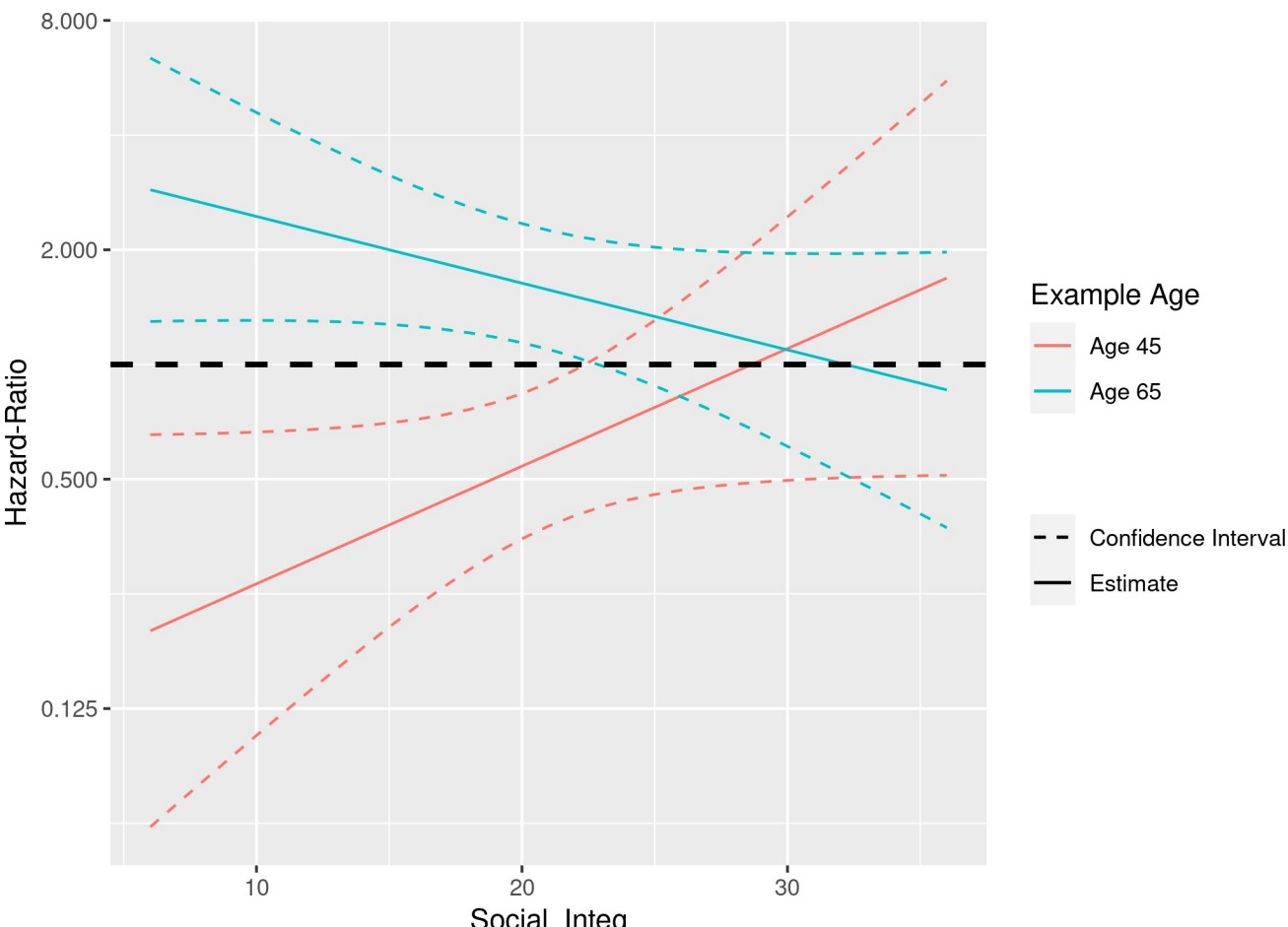

**Fig 2. Age and lack of social integration among older women with CAD.** There is an interaction between social integration and age. The predicted risk ratio for mortality increases with the degree of social isolation in older women (65 years) and decreases in younger women (45 years). This result could be plausible, as a lack of social integration could be an additional strain for older women.

 b. Work stress (HR 1.0, 95% CI 0.96–1.03) and homemaker status (HR 1.01, 95% CI 0.69–1.47) did not predict all-cause mortality after 26 years.

2. Behavioral risk factors

 a. Practising moderate exercise compared to a sedentary lifestyle (WHO criteria) was highly predictive for survival (HR 0.54, 95% CI 0.37–0.79, p = 0.002) (Fig 3A). In the sub-analysis of the women with regular-intensity exercise (n = 13), we found they were younger, smoked less, and had lower blood glucose than the sedentary and moderate exercise group but died earlier than patients in the moderate exercise group (Table 1). PA (posture) during 24 h ECG showed the same result as the WHO criteria but did not reach predictor quality for survival (HR 0.92, 95% CI 0.84–1.01, p = 0.073) (Table 3A).

 b. Smoking after 26 years was a significant predictor of all-cause mortality (HR 1.56, 95% CI 1.03–2.36) (Table 3A and Fig 3B).

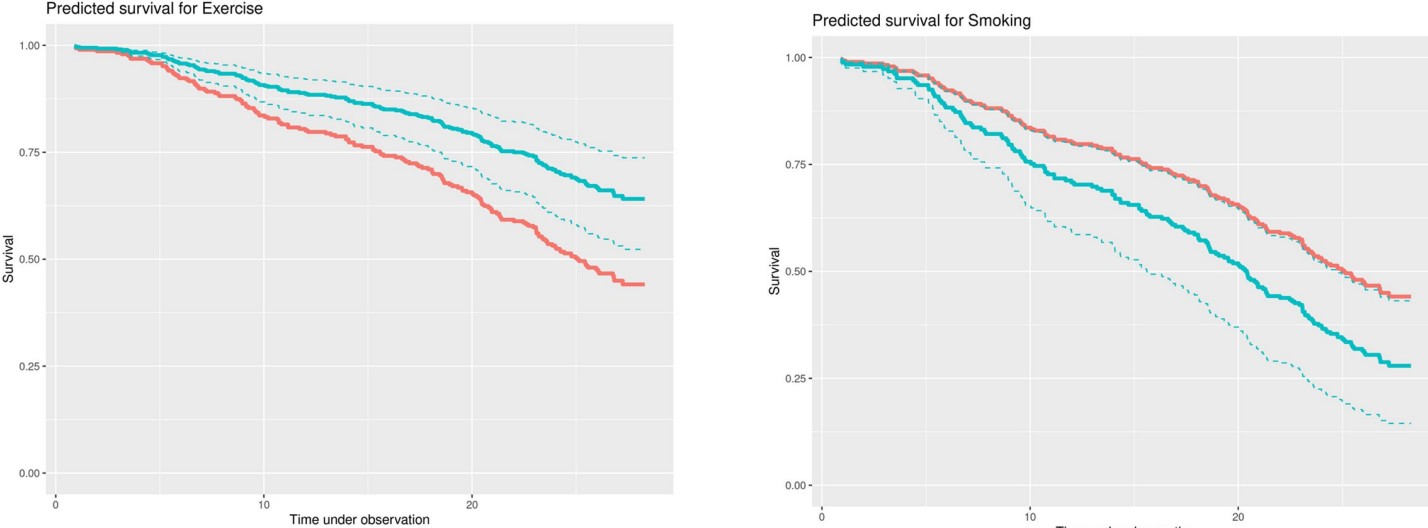

**Fig 3. Predicting survival for exercise and non-smoking in women with CAD.** HRs do not allow quantifying effects on survival directly, therefore, predicted survival curves are displayed for a selection of covariates from the final model. For all plots, orange codes the baseline survival curve, cyan codes survival for relevant values of the covariate considered. **a) Exercise.** Moderate Exercise seems to be beneficial, showing an increase in survival of 20% appr. **b) Smoking.** Smokers might have a reduced chance of survival during follow up of up to 17% appr.

 c. Depression did not become a significant predictor for all-cause mortality (HR 1.01, 95% CI 0.94–1.08). Vital exhaustion (VE) did not reach the list of the most important 20 predictors in that model.

 d. Disturbed sleep was not a significant predictor of all-cause mortality (HR 1.06, 95% CI 0.71–1.59). Sleep quality was not included in the model.

As expected, other well-known baseline cardiovascular risk factors showed an HR between 1.18 and 2.85 and were significant predictors for all-cause mortality (Table 3A). DHEAS as an indicator of estrogen protection predicted survival (HR 0.67, 95% CI 0.46–0.99) in this model (Table 3A). A calculation of predictors of the 15-year follow-up showed similar results compared to the 26-year follow-up. Exercise (HR 0.42, 95% CI 0.24–0.74, p = 0.003), became significant and SI × age showed a statistical tendency (HR 0.99, 95% CI 0.98–1.0, p = 0.058). Interestingly at 15-year follow-up, current smoking was not significant.

## B. Predictors in age-matched healthy controls

This study made it possible to compare these 26-year follow-up predictors of all-cause mortality in CAD patients with predictors of mortality in age-matched healthy women (S1 Fig). A high proportionality (global score and standard deviation of normal-to-normal R-R interval (SDNN)), possibly due to the low number of mortality cases in this presumably healthy group, made this model less convincing than the patient model. Age was again a significant predictor (HR 1.13 95% CI 1.08–1.17; p = 0.001). Related to our hypotheses we found for SI in this group no associations with all-cause mortality as for work stress. Cardiac indicators possibly associated with behavioral stress SDNN (HR 0.98, 95% CI 0.96–1.0, p = 0.076) and HRV without dysrhythmias (HR 0.20, 95% CI 0.81–0.48; p = 0.001) emerged as predictors in the model. Related to behavioral risk factors, PA became again a signinificant predictor (HR 1.62 (95% CI 1.10–2.40; p = 0.016) (Fig 4). Current smoking (HR 1.59, 95% CI 0.93–2.72; p = 0.089) and depressive symptoms (HR 1.89, 95% CI 0.89–3.99; p = 0.097) were included in the predictive

## Predicted survival for Exercise

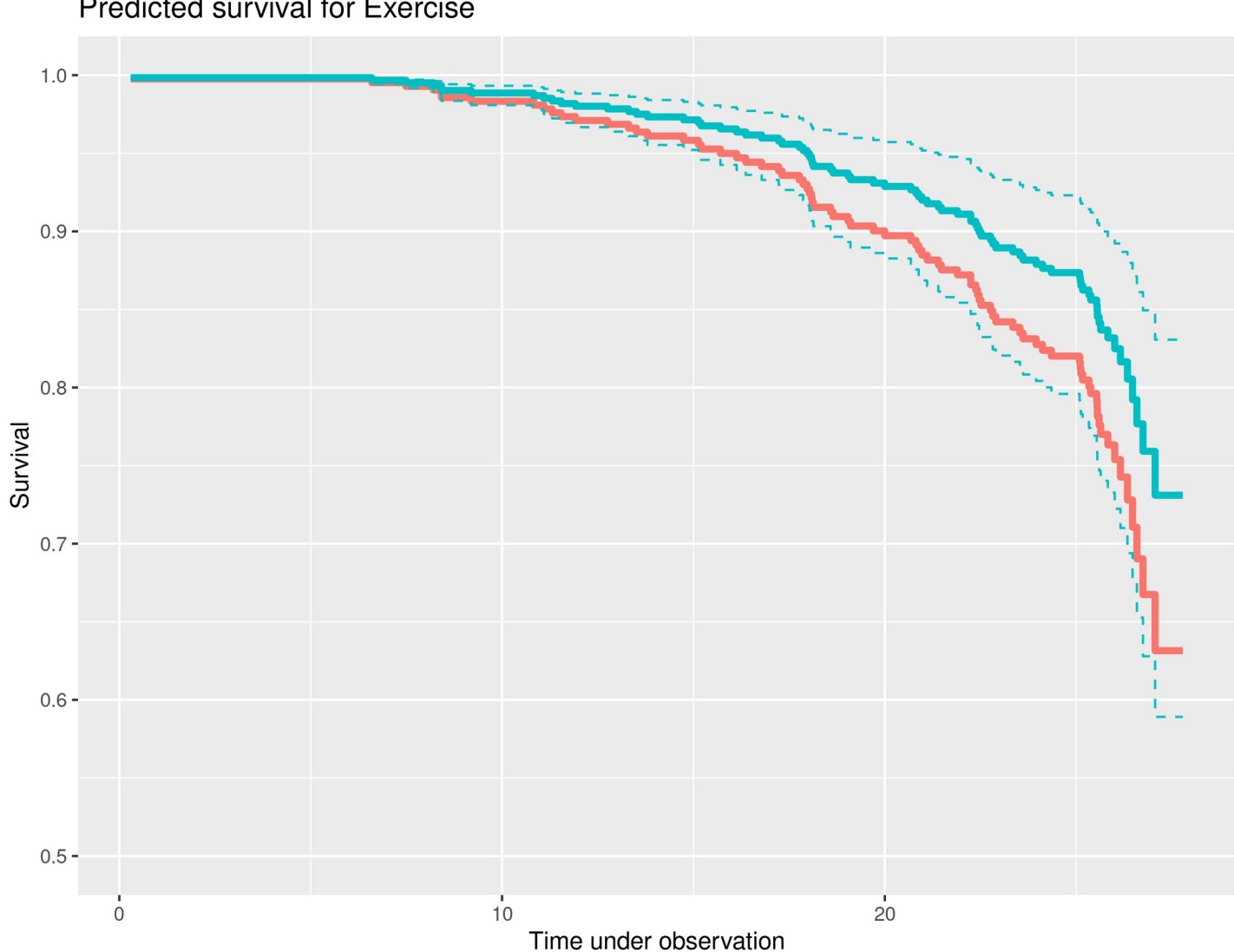

**Fig 4. Predicting survival for exercise in healthy women.** Moderate exercise seems to be beneficial, showing an increase in survival of 10% appr.

model but were not found to be significant (Table 4). Vital exhaustion and sleeping disturbances became not part of the model.

## Discussion

This prospective secondary analysis of the Stockholm Fem-Cor-Risk-Study, examined the predictive power of social strain and behavioral factors over 26 years. Few studies have demonstrated in a large female CAD cohort the association of social strain caused by social isolation, marital and work stress with all cause mortality over a long follow-up [4,8,11]. Social strain was associated with unhealthy lifestyle habits [12] which trigger psychobiological pathogenetic mechanisms of CAD progression and became predictors of all cause mortality in females with CAD [15,22,25–27]. The progression of CAD and outcome is associated with various gender-specific cardiac, endocrinological immunological and behavioral factors. In recent years,

**Table 4. Model for controls after 26 years.**

|  | Hazard-Ratio | 95% lower | 95% upper | z-value | p-value |
|---|---|---|---|---|---|
| Age | 1.126 | 1.079 | 1.174 | 5.492 | 0.000 |
| Practising exercise (Q-shape*) | **1.623** | **1.096** | **2.404** | **2.417** | **0.016** |
| Practising exercise (L-shape˚) | 1.258 | 0.683 | 2.316 | 0.736 | 0.462 |
| Current smokers | 1.593 | 0.932 | 2.724 | 1.701 | 0.089 |
| Former smokers | 1.292 | 0.730 | 2.287 | 0.879 | 0.379 |
| Depressive symptoms | 1.886 | 0.891 | 3.995 | 1.658 | 0.097 |
| CRP log | 1.307 | 0.967 | 1.767 | 1.740 | 0.082 |
| SDNN | 0.981 | 0.960 | 1.002 | -1.774 | 0.076 |
| HRV recordings >50% Evaluable | **0.196** | **0.081** | **0.476** | **-3.597** | **0.000** |

* Sedentary lifestyle and regular intensive exercise versus moderate exercise.

˚Sedentary lifestyle vs. moderate exercise vs. regular intensive exercise.

studies have shown that the manifestation and severity of CAD, as well as the mechanisms and cardiovascular risk factors, differ between the sexes [5,14,42].

We hypothesised that psychosocial strain factors such as social isolation, stress experience and behavioral risk factors have an impact on the life expectancy of female CAD patients and healthy controls over a long follow-up period.

For our calculations, we used a new statistical method based on machine learning [39] to include the full set of outcome variables. This very novel approach accounted for the intercorrelations of all variables in this study without selection. Other studies have used fewer variables, one variable to evaluate a hypothesis and only a limited set of control variables. The final model derived in our study allows predicting survival probabilities within each year of the follow-up period, which includes the complete set of baseline conditions.

## Female CAD patients

In this study on CAD females we could confirm that „few social contacts", „low physical exercise"and „smoking"remained significant in the model, which was able to predict all-cause mortality in CAD patients after 26 years.

1. We found a significant interaction of *social integration* with age. As hypothesized and confirmed in other studies [15,29], low social integration or social isolation in older female patients (>55 years) increased all-cause mortality. This was consistent with the Berlin Aging study. "Living alone" (present more in women than in men) and feelings of "emotional loneliness" in the significantly older study group (mean age 84.5 years) predicted total mortality [43]. A large review also came to similar conclusions [33]. Against our hypothesis and unexpected, in younger patients, a lack of social integration did not predict mortality. Instead an unfavourable outcome in the long-term follow-up was predicted by frequent social interactions. Whether frequent social activities are dangerous to a special form of CAD in younger women [44] after an acute coronary event or a gender-specific aspect of the female reproductive system are involved, we could partly clarify looking on DHEAS, a precursor of oestrogen associated with the ageing process and in our study an important predictor for survival. Consistent with the finding that low oestrogen levels do not protect female CAD patients [44], low DHEAS and low social integration led to high mortality in patients over 55 years of age. Patients ≤55 years with low social integration and presumably higher DHEAS levels than in patients >55 years, had low mortality. The

buffer hypothesis of the primary objective in the FemCorRisk study that parts of the reproductive system protect against social strain, seems to be supported by this finding.

In contrast to a review [1], another long-term follow-up study in the USA [4] and the finding of an association between low social education and coronary stenosis narrowing in the Fem.Cor-Risk angiography study [12], low education, possibly an indicator of social strain, did not predict all-cause mortality in our model. In the individual analysis, we found no effect in healthy controls and a very small effect contrary to our expectation among patients: patients from lower social class showed marginally lower mortality. This finding is in contradiction to a review on Nordic countries (including Sweden) that demonstrated relatively high levels of mortality inequalities in Sweden, although the socio-economic differences in the Swedish population were not as pronounced as in other countries [45]. In addition, effects differ between men and women and influences from other risk factors [46] that are related to the influence of education on mortality in different countries. Indeed, the importance of the educational level seems to diminish with age among women and is negligible at the age of 85 [47] Another possible explanation could be that the cut-off in education was set between mandatory education and upper-secondary school. Statistics indicate that the difference in life-length is more pronounced between those with upper-secondary education or lower and higher education. Also, the finding that low education does not predict all-cause mortality in these Swedish CAD patients could be related to several influencing factors: The data of CAD patients admitted to any of the 10 coronary care units in Stockholm for an acute coronary event and the healthy controls included in this study were representative for Stockholm County and may differ to the general population. Although, while this study included the major confounders, potential effects of unmeasured characteristics and direct and indirect selection mechanisms may also play a role.

Behavioral risk factors:

a. The exploratory analysis demonstrated that *PA* was associated significantly with survival in female CAD patients 26 years after the first cardiac event. In the 9-year follow-up of this sample, a sedentary lifestyle was already a significant predictor of total mortality [27]. In several studies in male and female CAD patients and in epidemiological surveys, a sedentary life style was at risk for further cardiovascular events [28]. A large meta-analysis in women demonstrated the benefits of PA in relation to all-cause mortality [48]. The 26 years long-term follow-up in 287 women with CAD could confirm the fact that PA is beneficial compared to a sedentary life style. PA was associated with a significantly reduced risk of overall mortality in both men and women (HR 0.72, 95% CI 0.62 to 0.84 and HR 0.81, 95% CI 0.72 to 0.92, respectively[48]. When analysing in this study [49] the dose–response association of continuous minutes of weighted moderate to vigorous physical activity per day at baseline, authors observed a reduction in the risk of incident CHD which was specially marked between 20 and 40 min of physical activity per day. In our study, moderate exercise was more beneficial than in the very small group of CAD patients who did regular-intensity exercise. A high dose of "physical activity" in women with CAD could be unfavorable. Despite the low sample size in this group this result is interesting. Then there is an ongoing discussion about the proper amount of exercise training in cardiac rehabilitation [28]. Our results argue for cautious use of this preventive measure in women with AMI or UA.

b. *Smoking* showed a high predictive power for all-cause mortality after 26 years in these female CAD patients. Smoking after 15 years was not a significant predictor. These differences in probabilities in 15 and 26 years of follow-up indicate a higher power of this predictor with the length of the observation time. Current smokers had a higher risk than former

smokers, but we did not calculate the packages/year of cigarettes in our study. As in a 20-year follow-up [50], the severity of smoking and the level of left ventricular dysfunction seemed to influence the proportion of all-cause mortality.

Due to the length of time since the first cardiovascular event, other hypothesized social strain and behavioral risk factors did not reach significant predictor qualities. However, several non-significant HRs with a low 95% confidence interval and a statistical trend point to the direction, strength, and precision a variable might predict in this explorative analysis:

*Stress experience*: After selecting only married or patients with a partner, this follow-up found that *marital stress* was included in a significant model but did not reach statistical significance. So the predictor qualities of marital stress diminished during long-term follow-up. An association between *work strain* and 26-year mortality in CAD patients was weak. Work strain was included in the model as a hypothesized predictor but did not show significant predictor qualities. Because patients were enrolled in this study with a median age of 56 years, most patients had already retired after 26 years. In former studies with shorter follow-up times, work strain had a higher impact [6,7,51]. In a national representative cohort of French employees, the associations of current exposure to job strain and mortality were found to be higher among women (HR = 1.15 (95% CI: 1.06–1.25) [52].

*Depression* entered the model as a possible predictor of mortality. Many studies reported depression as an important predictor of MACE or atherosclerotic coronary progression [15,18,19]. The low HR value of the prediction could be due to a decreasing effect over time or the kind of measurement: In our study, depressive symptoms were measured by the Pearlin scale [37] and not by an instrument to detect diagnoses of severe depression. The somatic depression subtype measured with this scale did not become a part of the model. One review found that only some depressive symptoms could predict mortality in women 56–69 and > 70 years of age [53]. In long-term follow-up studies, depression was a predictor in male but not female CAD patients [54]. In our study with unselected female CAD patients, depressive symptoms became not significant as a predictor for 26-year mortality. *VE* was not chosen for the final model in our study. There could be a gender effect; our study focused on women, other studies mostly examined males and only in a smaller part female [54]. The low prediction quality could also be due to a diminishing effect over time.

*Sleep disturbances* influenced the development of cardiovascular diseases in CAD patients [26] and in the healthy Fem-Cor-Risk control sample after 9 years [32]. Our hypothesis that sleep disturbances influence mortality at very long-term follow-up could not be confirmed. Additional collection of sleep data during a longitudinal study, as well as the inclusion of other variables (depression, fatigue, snoring, work strain), would enhance the quality of the predictor analysis.

**Psychobiological pathogenetic mechanisms.** *HRV*, *PA and risk factors*: In our female CAD patients and controls, it could be shown at baseline that sedentary lifestyle was associated with HRV (SDNN and LF/HF ratio), BMI, smoking, and depression [23,31]. Significant predictors of all-cause mortality within a median follow-up of 9 years were SDNN index, total power, VLF, LF and HF power [22]. The results were essentially the same when cardiovascular mortality was used as end-point, but the HRV parameters were stronger predictors of mortality in the first 5 years following the index event [22]. This could be the reason why HRV was not a predictor in our 26-year prediction model. In the Pregnancy Outcomes and Community Health Study, total and leisure-time PA were associated with favorable lnSDNN and lnRMSSD. Sedentary behavior during leisure time was associated with an unfavourable lnRMSSD (B = -0.041, [p = 0.042]) only in women [55]. All-cause mortality over a long follow-up period associated with moderate to vigorous PA could not be investigated in this study.

*LV dysfunction*, *Killip/NYHA class*, *glucose*, *and age* were further predictors with increasing power compared to the shorter follow-up of the Fem-Cor-Risk-study [3] and other studies [1,2]. These comorbidities also interact with behavioral factors in predicting mortality during follow-up. -Other behavior-related factors such as eating habits, waist-to-hip ratio, cholesterol and triglycerides were not included in the Cox boost prediction model, although some baseline data differed between surviving and deceased patients.

Compared to a review that examined eight risk factors related to the influence of education on mortality in different healthy and sick samples and in different countries [46], we were able to confirm in our study of female CAD patients that "few social contacts", "low physical activity" and "smoking" remained significant in the model that predicted all-cause mortality in CHD patients after 26 years. "Social class"(low income, education), „high body weight", „high alcohol consumption"and „high energy consumption"were not included in the model. The variable „father with a manual occupation"we had not included in our measurement. These and other various remaining causes of death demonstrate the complexity of all-cause mortality and we have to accept diverse aspects of unobserved heterogeneity in our analyses.

## Healthy female controls

In the study on healthy females we could confirm that „low physical exercise"remained significant in the model, which was able to predict all-cause mortality in healthy controls after 26 years. Smoking and depressive symptoms became part of the model, social class (low income, education), few social contacts, high body weight, and high alcohol consumption were not included in the model [46]. Due to the small number of deaths and the large deviations from the proportional hazard assumption of the model in the healthy controls after the 26-year follow-up period, the results should be treated with caution.

Some results in controls are in good agreement with the CAD-patients:
Behavioral risk factors

a. *PA*: In concordance to the results in CAD patients, a sedentary lifestyle or regular intensive exercise versus moderate activity was a significant behavioral predictor for all-cause mortality in healthy women. In a meta-analysis of 59 studies in women [48], PA was associated with a lower risk for all-cause mortality (HR 0.71 (95% CI 0.65–0.78).

b. The prediction of *current smoking* after 26-year follow-up was higher than former smoking, but both became not significant. In a meta-analysis [48] in healthy women, current smoking showed a higher risk (HR of 2.22 (95% CI 1.92–2.57)) for all-cause mortality. The severity of smoking seems to play a role in this context.

c. In the healthy control group, *depression* also appeared as a predictor for mortality in the model (HR 1.89, 95% CI 0.89–4.0; p = 0.097)), but this was not significant. In a sample of healthy males, depression predicted fatal ischemic coronary disease events after 20.9 years follow-up (adjusted HR 1.07, 95% CI 0.99 to 1.15; p = 0.08) [54].

**Psychobiological pathogenetic mechanisms.**   In the control group, we were not able to study cardiovascular parameters such as LV dysfunction or Killip/NYHA class or pregnancy complications, but HRV and biochemical measures.

Interestingly, in healthy women, we discovered two HRV variables as predictors of all-cause mortality that could be triggered by social strain or unhealthy lifestyle habits [12]. At baseline, different HRV variables were associated with social isolation. Depressive symptoms were related to the LF/HF ratio [31]. In this follow-up, decreased HRV variability (SDNN)

became part of the model and showed a tendency of risk with a small overlap with no effect levels. It could be an indicator for long term mortality in healthy women [22–24]. Women with complete HRV records compared to women with incomplete HRV records, possibly caused by dysrhythmias, showed a high survival.

A diagnosis of cancer in the 9-year follow-up of healthy controls [32] in this study predicted higher mortality in the subsequent follow-up period up to 26 years. Pregnancy complications, which we did not study, increase the risk of cardiovascular disease later in life: For preterm birth, the adjusted HR for cardiovascular mortality was 1.84 (95% CI, 1.38–2.44) [56]. Age at baseline had a similar effect on survival in both groups.

In presumably healthy women, a sedentary lifestyle poses a significant risk for all-cause mortality. Current smoking and depressive symptoms appear to be associated with long-term mortality. Reduced HRV and HRV irregularities are likely to be important psychobiological pathogenetic mechanisms in this context.

In women with CAD, a long-neglected patient group [5], social isolation, a low level of physical activity and current smoking have proved to be socio and behavioral predictors for an unfavorable outcome. The situation of male CAD patients will probably vary [57,58]. The influence of some other predictors studied and not included in the model may vary during the follow-up period: It is possible that the association of some behavioral factors related to MACE is time limited [22]: After 7 years, the proportion of deaths is highest in patients compared to the control group with a factor of 9.0 after 7 years. Later, this difference decreased to a factor of 3.4 after 15 years and to a factor of 1.6 after 26 years, showing that influences other than the original cardiac ones became more important.

Presumably work stress, depression, VE and sleep disturbances measured at baseline had a short or medium-term effect. PA and smoking could have a longer-lasting effect if the behavior at the beginning of the study is maintained during the follow-up period.

Moreover, the variables could influence each other: for example, we were able to demonstrate a significantly different influence of age and social integration on all-cause mortality. A full disentangling of the influence and sustainability of single variables as strain, depression, and sleep disturbances on mortality could not be achieved without several intermediate checks for conditions during the observation period and information about cause of death.

The results of both regression analyses after 26 years of follow-up can be summarised as follows: Socially isolated older women with CAD had higher all-cause mortality. Women with CAD and healthy women with sedentary lifestyles as well as women with cardiovascular disease and healthy women with smoking habits had higher all-cause mortality within 26 years. In addition, we were able to select different behavioral predictors that came into the model but were not significant. However, the hazard ratio and the confidence interval show the importance of these factors for the prediction of all-cause mortality.

## Limitations

a. The basis of this examination was the complex data pool of the Fem-Cor-Risk study and all-cause mortality data. A statistically pre-planned analysis and different surveys within the follow-up period with time-dependent covariate information would have strengthened their explanatory power. When examining survival time for all-cause mortality in our cohorts, we received limited information about cardiac death, recurrent MI or PCI during the first years of follow-up [8,22,27]. Assuming we had the complete information, a much more appropriate method of analysis would have been possible. For healthy controls several cardiac markers of CAD were not available We opted to reject the idea of a common model for analyzing risk factors, as controls and patients models are different.

b. For the calculation in this exploratory study, we used an established machine-learning-based statistical method (Cox Boost multiple regression on survival-times [39]) to include the complete set of baseline variables. We did not provide formal testing between patients who survived and died, avoiding misinterpretation of apparent relations.

c. In the first year, the severity of CAD (left main stenosis and 3-vessel disease) is an important risk factor for cardiovascular events [59]). The impact of social and behavioral cardiovascular risk factors in the Fem-Cor-Risk sample was presented after three to five years [5,8,27,30] in a situation where patients faced acute disease limitation. Due to life changes and intervening influences in the long follow-up period, the impact of these factors became blurred.

d. This study started with the morphological knowledge about coronary arteries at that time. Today, endothelial dysfunction, reduced microcirculation and myocardial perfusion measured by SPECT, or coronary computed tomography angiography [60] can specify more clearly patients with and without obstructive CAD on different vessels [61]. We could not show whether most of these specifications or the effect of new treatments are related to the behavioral factors studied here.

e. Psychological and sociological rating instruments known to be potentially important for prognosis were used at baseline. New psychological constructs, for example, anxiety, negative affectivity or substance abuse, were not applied but might have more predictive power.

Our results show at first time in female CAD patients strong predictors of all-cause mortality at 26 years. These results should be replicated and extended by longitudinal studies during the follow-up period and by collecting data on cardiovascular events and cardiac mortality. Future studies will show whether new cardiac or behavioral diagnostic procedures and treatments will elicit additional predictors of survival.

We were able to present with the support of a novel statistical method based on machine learning new findings on the course of CAD women and their healthy controls and open the window for long-term socio-behavioral studies on cardiac disease progression. This could initiate new research on long-term interactions between social strain, behavioral, cardiac and endocrinological factors, including the identification of their psychobiological pathogenetic mechanisms.

## Conclusions

This study began in 1991–1994 with a comprehensive baseline survey of women with cardiovascular disease and is unique in providing almost complete data on all-cause mortality over 26 years in patients (97.9%) and healthy controls (99.7%). It successfully demonstrated a new statistical method of performing survival analyses based on individual survival times in combination with modern regression techniques. The data analysis provided sufficient evidence for the importance of behavioral risk factors for all-cause mortality in this cohort of female CAD patients: social isolation in elderly patients, lack of exercise and smoking play an important role alongside known risk factors such as age, LV dysfunction and diabetes. In addition to the usual cardiological treatment of CAD, targeted behavioral therapy intervention to improve social integration is desirable in these patients. In women with cardiovascular disease with smoking habits, smoking cessation and other preventive measures are needed to reduce all-cause mortality within 26 years. In healthy controls as in CAD patients, PA was the most important predictor of survival. In women with CAD and in presumably healthy women with sedentary lifestyles, programmes to increase PA are likely to be helpful. Developing

appropriate behaviors and regulating social relationships as part of an individual treatment strategy or targeted intervention could make prevention programmes more effective and help healthy and sick women live longer and better lives.

## Supporting information

**S1 Fig. Distribution of age.** The density plot of the matched ages shows nearly perfect agreement between patients (in red) and controls (in blue). Interesting is the maybe mix of younger and older patients, giving a hint to a heterogeneity in the patients. The curves in Fig 1 have to be interpreted with this age distribution in mind.
(TIF)

**S2 Fig. Social Integration and age at baseline: Individual survival of CAD patients during 26-years of follow-up.** X-axis and the model variable present the age at baseline. The graphic compares the prediction of our model and the mortality data obtained from the study. The two black lines are reference lines, where the hazard ratio (HR) lines (red and light blue) appear as stripes in the background. HR for mortality values >1 is dark red, HR values <1 are shown in blue. The events appear as plots of squares (censored alive at 15-year follow-up), circles (censored alive at 26-year follow-up), or triangles (dead at 26 year-follow-up); larger shapes represent more individuals with the same age and the same social integration score.
(TIF)

**S1 Table. Social integration, received stress and behavioral risk factor as predictors of increasing coronary stenosis, recurrent cardiac events, cardiac and all-cause mortality—short-term results within the Female Coronary Risk Study.**
(PDF)

**S2 Table. Survival.** Kaplan Meier estimates of survival probablility per year of follow up for 286 CAD patients and 299 healthy controls, expressing variability by this coding.
(PDF)

**S3 Table. Test of proportionality in patients after 26 years follow up.**
(PDF)

**S1 File. Further descriptions of the study design, baseline examinations and non-significant predictors (in bold) examined in the Stockholm Study of Coronary Risk in Women (1991–1994) at 26-year follow-up.**
(PDF)

**S2 File. Additional statistical description.**
(PDF)

**S3 File. Details of the statistical analysis.**
(PDF)

## Acknowledgments

We thank the members of the FemCorRisk study (PI: K.O-G), especially Myriam van Rooijen-Horsten, PhD, (Manager at Centraal Bureau voor de Statistiek, Den Haag, Netherlands; m. vanrooijen@cbs.nl), Sarah Wamala-Andersson,D.M.Sc. (Professor at Mälardalen University; swamala8@gmail.com), May Blom,PhD (former Manager at Stockholm County Council, therapeut, group leader; mayblom1@gmail.com), Murray A Mittleman, (Professor of Epidemiology; Dept of Epidemiology, Harvard School of Public Health; mmittelm@hsph.harvard.edu)

and Karin Schenck-Gustafsson MD. (Department of Medicine, Solna, affiliated to teaching/ tutoring; karin.schenck-gustafsson@ki.se). We acknowledge the support of the Stress Research Institute, Stockholm University, in conducting this follow-up and thank Ursula Rauch-Kröhnert, Cardiology, Charité Campus Benjamin Franklin (Ursula.Rauch@charite.de), for her helpful comments on the manuscript.

## Author Contributions

**Conceptualization:** Hans-Christian Deter, Kristina Orth-Gomér.

**Data curation:** Hans-Christian Deter, Constanze Leineweber, Lukas Lohse, Kristina Orth-Gomér.

**Formal analysis:** Reinhard Meister, Göran Kecklund, Lukas Lohse.

**Funding acquisition:** Kristina Orth-Gomér.

**Investigation:** Hans-Christian Deter, Kristina Orth-Gomér.

**Methodology:** Reinhard Meister.

**Project administration:** Hans-Christian Deter, Constanze Leineweber, Kristina Orth-Gomér.

**Software:** Reinhard Meister, Lukas Lohse.

**Supervision:** Constanze Leineweber, Göran Kecklund, Kristina Orth-Gomér.

**Validation:** Hans-Christian Deter, Reinhard Meister, Göran Kecklund, Lukas Lohse.

**Visualization:** Lukas Lohse.

**Writing – original draft:** Hans-Christian Deter.

**Writing – review & editing:** Reinhard Meister, Constanze Leineweber, Göran Kecklund.

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
