## [Decision Letter · Decision Letter 0]

8 Aug 2022

PONE-D-22-07656Behavioral factors predict all-cause mortality in female coro-nary patients and healthy controls over 26 years

 – a prospective secondary analysis of

      the Stockholm Female Coronary Risk StudyPLOS ONE

Dear Dr. Hans-Christian Deter,

Thank you for submitting your manuscript to PLOS ONE. After careful consideration, we feel that it has merit but does not fully meet PLOS ONE’s publication criteria as it currently stands. Therefore, we invite you to submit a revised version of the manuscript that addresses the points raised during the review process.

We look forward to receiving your revised manuscript.

Kind regards,

Ricardas Radisauskas

Academic Editor

PLOS ONE

https://journals.plos.org/plosone/s/file?id=ba62/PLOSOne_formatting_sample_title_authors_affiliations.pdf".

a) Did participants provide their written or verbal informed consent to participate in this study?

3. Please include a complete copy of PLOS’ questionnaire on inclusivity in global research in your revised manuscript. Our policy for research in this area aims to improve transparency in the reporting of research performed outside of researchers’ own country or community. The policy applies to researchers who have travelled to a different country to conduct research, research with Indigenous populations or their lands, and research on cultural artefacts. The questionnaire can also be requested at the journal’s discretion for any other submissions, even if these conditions are not met.  Please find more information on the policy and a link to download a blank copy of the questionnaire here: https://journals.plos.org/plosone/s/best-practices-in-research-reporting. Please upload a completed version of your questionnaire as Supporting Information when you resubmit your manuscript.

4. We note that the grant information you provided in the ‘Funding Information’ and ‘Financial Disclosure’ sections do not match. When you resubmit, please ensure that you provide the correct grant numbers for the awards you received for your study in the ‘Funding Information’ section.

5. Please note that in order to use the direct billing option the corresponding author must be affiliated with the chosen institute. Please either amend your manuscript to change the affiliation or corresponding author, or email us at plosone@plos.org with a request to remove this option.

6. One of the noted authors is a group or consortium [Fem-Cor-Risk Study group]. In addition to naming the author group, please list the individual authors and affiliations within this group in the acknowledgments section of your manuscript. Please also indicate clearly a lead author for this group along with a contact email address.

Additional Editor Comments:

Dear manuscript authors,

Thank you for the forwarded manuscript.

Currently, the manuscript has considerable major shortcomings and therefore must be corrected and supplemented according to the comments of the reviewers.

Reviewers' comments:

Reviewer's Responses to Questions

**Comments to the Author**

1. Is the manuscript technically sound, and do the data support the conclusions?

Reviewer #1: Yes

Reviewer #2: Partly

2. Has the statistical analysis been performed appropriately and rigorously? 

Reviewer #1: Yes

Reviewer #2: Yes

3. Have the authors made all data underlying the findings in their manuscript fully available?

Reviewer #1: Yes

Reviewer #2: No

4. Is the manuscript presented in an intelligible fashion and written in standard English?

Reviewer #1: Yes

Reviewer #2: Yes

5. Review Comments to the Author

Reviewer #1: Thank you for the great work. This manuscript is an original article. In my opinion, the Introduction section is too long. The “Methods” section is written very well. There is a clear presentation of behavioral risk factors and data analysis. The results section: Table 1 there are presented the results of other authors. In my opinion, this table should be described in the Introduction section and added as a supplementary table or in the Discussion section, but also as a supplementary table. The discussion section is written very well, very deeply. The results are properly interpreted and there is a comparison with other studies provided.

Reviewer #2: The manuscript examines a hypothesis whether behavioral factors may predict all-cause mortality in female coronary patients and healthy controls. More specifically, the authors assume that since social isolation, marital stress, sedentary lifestyle and depression predicted CAD progression and outcome within 3 to 5 years, the same factors these behavioral factors would still be associated with all-cause mortality in female patients after 26 years. The study concludes that “CAD patients with adequate social integration, who do not smoke and are physically active, have a favorable long-term prognosis.” This is a potentially important contribution, but (in my view) the authors have to put more efforts in highlighting the novelty and impact of the study. One of the main advantages (and novelty) of the study concerns the application of the new machine-learning approach (the Cox Boost concept). However, the basic application procedure and its advantages/added value should be better explained for a general readership. Such explanations like “tuning of the fitting algorithm in a pragmatic way” are not sufficient. The discussion section should be more condensed and structured and more focusing on novelty and impact of the core findings. The result that low education does not predict all-cause mortality in Sweden is puzzling and should be better explained. At the whole population level, it has been shown that despite strong social security, the Nordic countries (including Sweden) show relatively high levels of mortality inequalities. This unexpected pattern is even called the “Nordic paradox” (see Mackenbach JP. Nordic paradox, Southern miracle, Eastern disaster: persistence of inequalities in mortality in Europe. Eur J Public Health. 2017 Oct 1;27(suppl_4):14-17. doi: 10.1093/eurpub/ckx160. PMID: 29028239.). It is clear that the data used in this study is not representative for the whole Sweden, therefore authors should elaborate more about the peculiarities of the study population (e.g. differences from general population, etc.). Although the authors include the major confounders, the potential effects of unmeasured characteristics (unobserved heterogeneity) and direct and indirect selection mechanisms may also play a role. These issues are potentially very important in the context the complexity (in terms of various remaining causes of death) of all-cause mortality and should be discussed in the discussion section. Finally, the authors should elaborate more on the policy and research implications (besides one sentence in the conclusion section).

6. PLOS authors have the option to publish the peer review history of their article (what does this mean?). If published, this will include your full peer review and any attached files.

Reviewer #1: No

Reviewer #2: No

---

## [Author Response · Author response to Decision Letter 0]

21 Sep 2022

Additional Editor Comments:

Dear manuscript authors,

Thank you for the forwarded manuscript.

Currently, the manuscript has considerable major shortcomings and therefore must be corrected and supplemented according to the comments of the reviewers.

We have tried to correct and supplemented shortcomings according to the comments of the reviewers

5. Review Comments to the Author

Reviewer #1: Thank you for the great work. This manuscript is an original article. In my opinion, the Introduction section is too long. 

We have shortened the introduction section (pg.3 and 4)

The “Methods” section is written very well. There is a clear presentation of behavioral risk factors and data analysis. The results section: Table 1 there are presented the results of other authors. In my opinion, this table should be described in the Introduction section and added as a supplementary table or in the Discussion section, but also as a supplementary table. 

We have described the results of Table 1 in the Introduction section (pg.4, l.10) and added Table 1 in the supplement as S1 Table.

The discussion section is written very well, very deeply. The results are properly interpreted and there is a comparison with other studies provided.

Reviewer #2: The manuscript examines a hypothesis whether behavioral factors may predict all-cause mortality in female coronary patients and healthy controls. More specifically, the authors assume that since social isolation, marital stress, sedentary lifestyle and depression predicted CAD progression and outcome within 3 to 5 years, the same factors these behavioral factors would still be associated with all-cause mortality in female patients after 26 years. The study concludes that “CAD patients with adequate social integration, who do not smoke and are physically active, have a favorable long-term prognosis.” This is a potentially important contribution, but (in my view) the authors have to put more efforts in highlighting the novelty and impact of the study. 

We have put more efforts in highlighting the novelty and the impact of the study in the discussion section and conclusion: (pg 20, line 3 and 1 from bottom; pg.21, l.4; pg 25, l.10 from bottom; pg.27, l.18; pg.28, l 9; pg. 30, line 2)

One of the main advantages (and novelty) of the study concerns the application of the new machine-learning approach (the Cox Boost concept). However, the basic application procedure and its advantages/added value should be better explained for a general readership. Such explanations like “tuning of the fitting algorithm in a pragmatic way” are not sufficient.

We corrected this sentence and presented a further description in the S 3 File. It states now (pg 10, l 11 from bottom):

and finally the tuning of the fitting algorithm required to avoid overfitting and spurious results due to leverages of the covariate data (S3).

And explained better the main advantages and the basic application of the new machine-learning approach for a general readership It states now (pg 20,l from bottom).:

For our calculations, we used a new statistical method based on machine learning (Cox boost multiple regression on survival; (39)) to include the full set of outcome variables. This very novel approach accounted for the intercorrelations of all variables in this study without selection. Other studies have used fewer variables, one variable to evaluate a hypothesis and only a limited set of control variables. The final model derived in our study allows predicting survival probabilities within each year of the follow-up period, which includes the complete set of baseline conditions. 

The discussion section should be more condensed and structured and more focusing on novelty and impact of the core findings. 

We have more condensed and structured the discussion section and focused on novelty and core findings, see above (pg 20, line 3; pg.21, l.4; pg 25, l.10 from bottom; pg.27, l.18; pg.29, l 10 from bottom; pg. 30 ,line 2)

The result that low education does not predict all-cause mortality in Sweden is puzzling and should be better explained. At the whole population level, it has been shown that despite strong social security, the Nordic countries (including Sweden) show relatively high levels of mortality inequalities. This unexpected pattern is even called the “Nordic paradox” (see Mackenbach JP. Nordic paradox, Southern miracle, Eastern disaster: persistence of inequalities in mortality in Europe. Eur J Public Health. 2017 Oct 1;27(suppl_4):14-17. doi: 10.1093/eurpub/ckx160. PMID: 29028239.). It is clear that the data used in this study is not representative for the whole Sweden, therefore authors should elaborate more about the peculiarities of the study population (e.g. differences from general population, etc.). Although the authors include the major confounders, the potential effects of unmeasured characteristics (unobserved heterogeneity) and direct and indirect selection mechanisms may also play a role. These issues are potentially very important in the context the complexity (in terms of various remaining causes of death) of all-cause mortality and should be discussed in the discussion section.

Thank you for discussing this important point. We have included the literature related to the „nordic paradox“ and „determinants of inequalities in life expectancy“ in the manuscript The levels of mortality inequalities in Sweden is higher in in men than in women and influencing factors (smoking, BMI) have been described as influencing factors on mortality beside educational inaequalities. In our model, education was not included, but smoking and physical activity. We have this issue and the aspect of „unobserved heterogeneity“ discussed more in detail in the discussion section. Pg 22, l 4 reads now:

“…low education, possibly an indicator of social strain, did not predict all-cause mortality in our model. In the individual analysis, we found no effect in healthy controls and a very small effect contrary to our expectation among patients: patients from lower social class showed marginally lower mortality. This finding is in contradiction to a review on Nordic countries (including Sweden) that demonstrated relatively high levels of mortality inequalities in Sweden, although the socio-economic differences in the Swedish population were not as pronounced as in other countries (45). In addition, effects differ between men and women and influences from other risk factors (46) that are related to the influence of education on mortality in different countries. Indeed, the importance of the educational level seems to diminish with age among women and is negligible at the age of 85 (47) Another possible explanation could be that the cut-off in education was set between mandatory education and upper-secondary school. Statistics indicate that the difference in life-length is more pronounced between those with upper-secondary education or lower and higher education. Also, the finding that low education does not predict all-cause mortality in these Swedish CAD patients could be related to several influencing factors: The data of CAD patients admitted to any of the 10 coronary care units in Stockholm for an acute coronary event and the healthy controls included in this study were representative for Stockholm County and may differ to the general population. Although, while this study included the major confounders, potential effects of unmeasured characteristics and direct and indirect selection mechanisms may also play a role.” 

 Finally, the authors should elaborate more on the policy and research implications (besides one sentence in the conclusion section).

A broader discussion on the policy and research implication was included in the conclusion (pg.30 l.3 and 13)

---

## [Editor Report · Decision Letter 1]

19 Oct 2022

Behavioral factors predict all-cause mortality in female coronary patients and healthy controls over 26 years

 – a prospective secondary analysis of the Stockholm Female Coronary Risk Study

PONE-D-22-07656R1

Dear Dr. Hans-Christian Deter,

We’re pleased to inform you that your manuscript has been judged scientifically suitable for publication and will be formally accepted for publication once it meets all outstanding technical requirements.

Kind regards,

Ricardas Radisauskas

Academic Editor

PLOS ONE
---

## [Editor Report · Acceptance letter]

3 Nov 2022

PONE-D-22-07656R1 

Behavioral factors predict all-cause mortality in female coronary patients and healthy controls over 26 years – a prospective secondary analysis of the Stockholm Female Coronary Risk Study 

Dear Dr. Deter:

I'm pleased to inform you that your manuscript has been deemed suitable for publication in PLOS ONE. Congratulations! Your manuscript is now with our production department. 

Kind regards, 

on behalf of

Professor Ricardas Radisauskas 

Academic Editor

PLOS ONE